# Molecular dynamics shows complex interplay and long-range effects of post-translational modifications in yeast protein interactions

**Nikolina Šoštarić** [1], **Vera van Noort** [1,2]*

**1** KU Leuven, Centre of Microbial and Plant Genetics, Leuven, Belgium, **2** Leiden University, Institute of Biology Leiden, Leiden, The Netherlands

* vera.vannoort@kuleuven.be

**Citation:** Šoštarić N, van Noort V (2021) Molecular dynamics shows complex interplay and long-range effects of post-translational modifications in yeast protein interactions. PLoS Comput Biol 17(5): e1008988. https://doi.org/10.1371/journal.pcbi.1008988

**Data Availability Statement:** No, some restriction will apply. All input topology and coordinate files, as well as scripts for MD runs and trajectory snapshots are available from the Zenodo database

## Abstract

Post-translational modifications (PTMs) play a vital, yet often overlooked role in the living cells through modulation of protein properties, such as localization and affinity towards their interactors, thereby enabling quick adaptation to changing environmental conditions. We have previously benchmarked a computational framework for the prediction of PTMs' effects on the stability of protein-protein interactions, which has molecular dynamics simulations followed by free energy calculations at its core. In the present work, we apply this framework to publicly available data on *Saccharomyces cerevisiae* protein structures and PTM sites, identified in both normal and stress conditions. We predict proteome-wide effects of acetylations and phosphorylations on protein-protein interactions and find that acetylations more frequently have locally stabilizing roles in protein interactions, while the opposite is true for phosphorylations. However, the overall impact of PTMs on protein-protein interactions is more complex than a simple sum of local changes caused by the introduction of PTMs and adds to our understanding of PTM cross-talk. We further use the obtained data to calculate the conformational changes brought about by PTMs. Finally, conservation of the analyzed PTM residues in orthologues shows that some predictions for yeast proteins will be mirrored to other organisms, including human. This work, therefore, contributes to our overall understanding of the modulation of the cellular protein interaction networks in yeast and beyond.

## Author summary

Proteins are a diverse set of biological molecules responsible for numerous functions within cells, such as obtaining energy from food or transport of small molecules, and many processes rely on interactions of specific proteins. Moreover, a single protein may acquire different roles depending on cellular requirements and as a response to changes in the environment. A commonly used way to quickly change protein's function or activity is by introducing small chemical modifications on specific locations within the protein. These modifications can cause the protein to interact in a more or less stable way with other proteins. We have previously developed a computational pipeline for predicting the

(http://doi.org/10.5281/zenodo.4099098). In addition, RMSD calculation files and structures of cluster representatives are given, as well as the Python script for automatic addition of post-translational modifications to protein structures. Other data underlying the results presented in the study are within the manuscript and its Supporting Information files. Raw MD output files (xtc trajectories) for all systems analysed in this study are currently not shared publicly because of their large size (a total of 3.7 TB). The authors will however readily share individual files with interested researchers (data can be requested by sending an email to vera.vannoort@kuleuven.be).

**Funding:** This work is supported by Onderzoeksraad, KU Leuven (KU Leuven Research Fund, https://www.kuleuven.be/onderzoek/ondersteuning/if) (to V.N.). N.S. is a doctoral fellow (1112318N) of The Research Foundation – Flanders (FWO, https://www.fwo.be). The funders had no role in study design, data collection and analysis, decision to publish, or preparation of the manuscript.

**Competing interests:** The authors have declared that no competing interests exist.

effect of modifications on interactions of proteins, and in this work we apply it to all yeast proteins with known structures. We find differences in effects on the binding for different types of modifications. Importantly, we demonstrate that the modifications far from the interaction interface also significantly contribute to binding due to their impact on protein's shape, which is often neglected by other methods. This work contributes to our understanding of the modulation of protein interactions in yeast due to modifications, while our widely applicable method will allow similar investigations in other organisms.

## Introduction

Since the first mentions of post-translational modifications (PTMs) in PubMed, dating to the 1940s, it has become increasingly clear that the numerous types of PTMs and their crosstalk[1] have indispensable roles in the functioning of organisms from all three domains of life. Additions of chemical moieties on designated amino acids are known to affect protein stability, activity and localization, as well as fine-tune the modified protein's binding to its interacting partners.[2,3] It has indeed been demonstrated for some types of PTMs that the interface located sites have higher conservation [4], suggesting that these PTMs might be more functionally relevant, as they can exert a direct influence on protein-protein interactions. In addition to these direct effects, PTMs are also known to allosterically impact the conformation of proteins, further altering their respective functions.[5].

Phosphorylation is arguably the most studied type of PTM. Discovered in the 1950s on phosphorylase extracted from the rabbit skeletal muscle [6], phosphorylation has since been associated with virtually all cellular processes [7], and up to 75% of the human proteome [8] is known to be affected by this modification. Similar to phosphorylation, lysine acetylation is another well-known PTM conserved in bacteria, archaea and eukaryotes, suggesting its ancient origin.[9] While it is still largely associated with its first discovery on histones [10], lysine acetylation was meanwhile found in many other proteins, e.g., tumor suppressor p53 [11], and is nowadays being studied at larger scales. For instance, lysine acetylation was found to be at least as abundant as phosphorylation in the bacterium *Mycoplasma pneumoniae*.[12] Thus far, acetylation has been implicated in a wide range of cellular roles, such as autophagy, cell cycle, and cytoskeleton organization, to mention just a few.[3].

The last two decades have witnessed a rapid increase in the number of identified PTMs for a number of organisms, primarily due to advancements in mass spectrometry (MS).[13,14] The current MS techniques allow not only the proteome-wide quantitative identification of PTMs but also their detection upon perturbations of cellular conditions, which aim to clarify the roles of these modifications in a dynamic and network signaling context. The abundant PTM data is stored in publicly available databases, which are often organism-, PTM- or amino acid-specific, for instance, the lysine-specific Protein Lysine Modification Database [15] or the yeast phosphorylation database phosphoGRID [16]. A collection of data from 30 such databases was unified within the dbPTM resource, with more than 900,000 experimentally confirmed and around 350,000 putative PTM sites of different types present in the latest release version of 2019 [17]. In addition to a large number of experimental and predicted sites, PTMs are also chemically very diverse: UniProt [18] report dated October 2020 listed 676 types of PTMs, 100 of which have been associated with the taxonomic range of Archaea, 319 with Bacteria, and 489 with Eukaryota.

Despite these impressive numbers of both types and identified PTM sites, what remains a bottleneck in this line of research is predicting the functional roles of these PTMs.[9,19] For

instance, as much as 95% of human phosphosites still do not have assigned functions.[20] The current situation in learning about the functional significance of PTMs has been compared to the early days of genomic sequencing, when the accompanying bioinformatics tools were lacking and had to be developed to convert the inflow of data into scientific knowledge.[9] To date, several tools and resources have emerged that try to tackle the problem of predicting PTM functions. PTMfunc [4] is one such resource, which offers the precomputed predictions of phosphorylation, acetylation and ubiquitination relevance in eukaryotic proteins, based on the PTM conservation within family domains. On the other hand, Mechismo web server [21] predicts if a PTM occurring on a specific position in a query protein has stabilizing, destabilizing, or no effect on binding with the respective interactors. These predictions rely on the interface interaction patterns, assuming the conformation remains identical before and after PTM addition. In addition to these, tools originally developed for different purposes have also been applied to predicting the roles of PTMs on binding. One such example is the FoldX empirical force field [22], developed with free energy calculation based on 3D structure as its core functionality, which has recently been used to estimate the effects of interface-located phosphorylations on binding [23]. Similar to Mechismo, predictions of FoldX are in practice limited to the PTMs located at the interaction interfaces; otherwise they predict that there is no effect of PTM introduction.

In our recent work [24], we aimed to surpass the drawbacks of the existing tools by developing a method that would take the dynamic nature of biomolecules into account and could accommodate for numerous PTM types present simultaneously in the structure, while not being limited by their interface location. This corresponds to a more realistic scenario, as it is known that a significant proportion of proteome carries multiple modifications.[12,25] Our proposed pipeline has molecular dynamics (MD) simulations at its core, followed by the free energy calculation using the Molecular Mechanics energies with Generalized Born and Surface Area continuum solvation method (MM/GBSA) at a set of protein conformational snapshots. [24] We have previously benchmarked the MD-MM/GBSA pipeline against FoldX [22] and Mechismo [26] using a set of 47 mammalian protein-protein complexes for which it was experimentally shown that phosphorylation causes either stabilization or destabilization of binding. The benchmarking showed that FoldX has the lowest accuracy; Mechismo can reach up to 75% accuracy at the cost of coverage, while MD-MM/GBSA has much higher coverage. After confirming that MD-MM/GBSA predictions are similar to those of other tools for stabilizing and better for destabilizing effects, yet offer more flexibility with respect to PTM type, number and location and account for dynamics, we further applied it to three multimeric protein complexes. Finally, we experimentally confirmed one of the obtained predictions by yeast-2 hybrid.

In the present work, we aimed to apply our MD-MM/GBSA workflow for predicting the effects of PTMs on binding stability to gain insight into the effects of acetylation and phosphorylation in the known protein-protein interactions of yeast *Saccharomyces cerevisiae* (Fig 1). To this end, we made use of the publicly available data on 3D structures of complexes, as well as the PTM data. This led to performing 440 MD simulations, with the possibility of mirroring some of the prediction results to other organisms due to PTM conservation in the orthologous proteins. Even though acetylations more frequently have locally stabilizing effects on binding, and phosphorylations destabilizing, in this work we demonstrate that the overall effect of co-occurring PTMs on the protein-protein binding is more complex than a sum of local changes, likely as a consequence of conformational changes. In addition, these local binding contributions do not appear to be correlated with the conservation of the PTM site, further emphasizing the complexity of predicting the effects of PTMs.

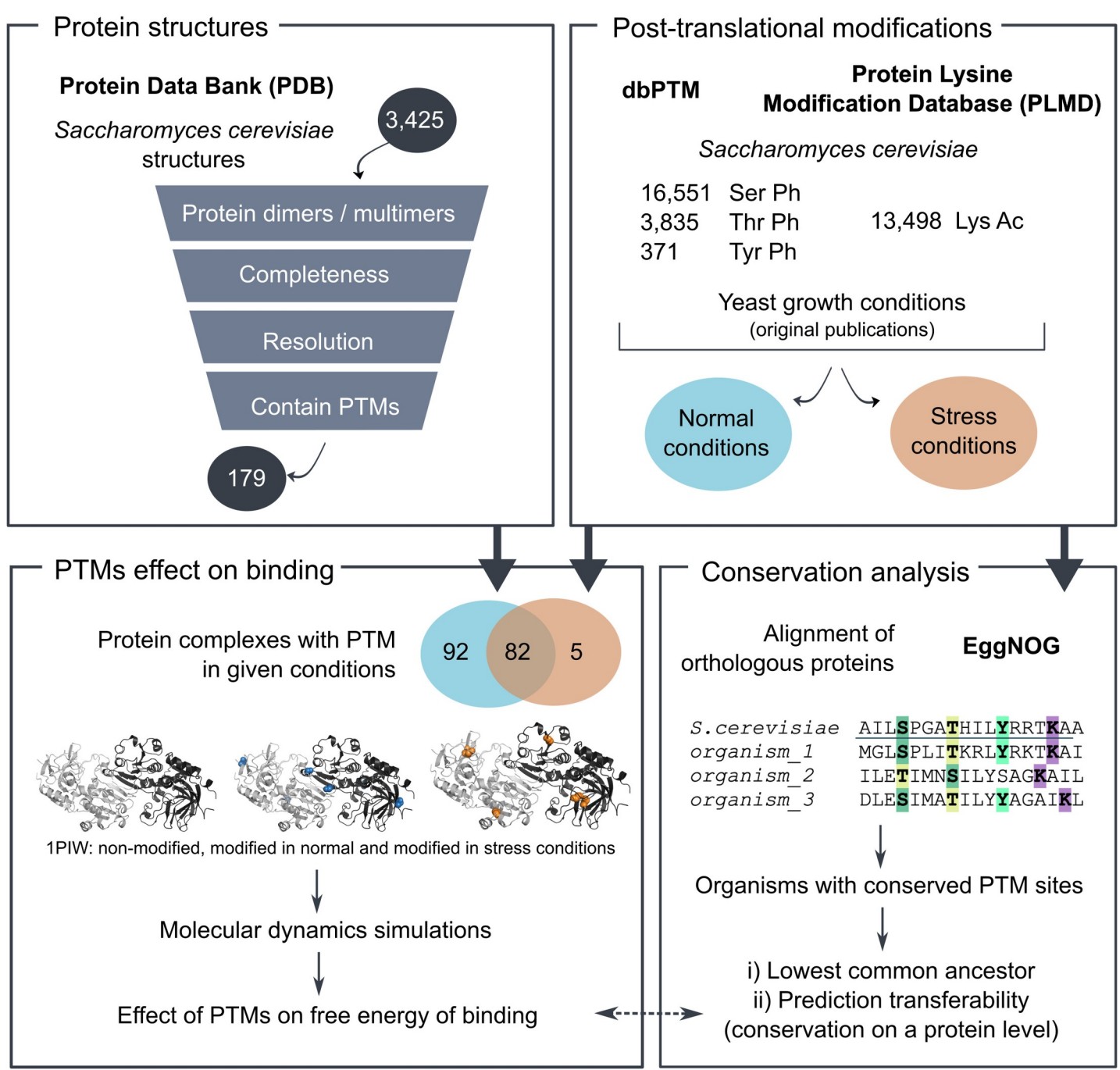

**Fig 1. Workflow overview.** The publicly available yeast protein structures were combined with PTM sites data in order to assess the effects of PTMs on protein binding. In addition, conservation analysis aided in identifying the subset of predictions that are transferable to protein-protein interactions in other organisms.

## Results

### Post-translational modifications of yeast proteins

The yeast protein complexes analyzed in this study were of different sizes, with the largest one having 6,386 residues, as well as of different multimeric states, ranging from dimers to a 28-mer (S1 Fig). Before modeling PTMs onto 3D structures of these protein complexes,

acetylation and phosphorylation sites of *Saccharomyces cerevisiae* retrieved from the public databases were separated based on the conditions (normal or stress) in which yeast was grown when each respective PTM has been identified. A total of 174 protein complexes in this study had PTMs found in normal conditions, which included the addition of 2,832 lysine acetylations and 289 phosphorylations on the respective complexes (S1 Table). On the contrary, only 87 complexes containing PTMs in stress conditions were identified, having no associated acetylations and a total of 605 phosphorylations (408 on serine, 177 threonine, and 20 on tyrosine residues). It is worth noting that these numbers contain some redundancy, as for example each subunit in a homomultimer typically contains the same set of PTMs. However, there are also exceptions, as in some cases the gaps in protein chains are not observed at the same locations in all subunits of a crystallized complex. Due to the same reason, one protein that occurs in multiple PDB structures, and therefore in different complexes, might also have slightly different PTMs added in each of those structures. Therefore, in the remainder of this section, "unique chain" is used to denote a single protein chain, with a unique UniProt ID, which is located within a specific PDB structure (i.e., a single chain from one row in the S1 Table).

A unique protein chain in our normal conditions dataset typically had several acetylations (15.8% of chains had one, 17.0% two, and 14.5% three acetylated lysines), though the number went above 20 in a few cases of hypermodified proteins (S2B Fig). Phosphorylations within the proteins in normal conditions were of a lower frequency–depending on the amino acid, between 80 and 98% of unique chains had no phosphorylation sites at all, and when they did, it was typically a single one (S2C–S2E Fig). Overall, 16.2% of unique chains had a total of one PTM, 15.8% had two, while the remaining had three or more modifications (S2A Fig). These numbers were, however, somewhat different for the proteins in stress conditions: while a majority of 70–90% of unique chains contained no phosphorylated threonines and tyrosines, only 14.7% had no serine phosphosites. Indeed, proteins in this dataset were commonly modified with up to three phosphoserines (50.5% had one, 15.6% had two, 8.3% had three modified residues). For those unique chains that did contain a threonine or a tyrosine phosphosite, it was most commonly only a single one (26.5% of chains had one phosphothreonine and 9.2% one phosphotyrosine). The lower occurrence of tyrosine phosphosites is not surprising, as this amino acid itself is represented in rather low numbers throughout yeast proteins.[27]

## Acetylation and phosphorylation have different local contributions to binding

Applying MD-MM/GBSA pipeline on post-translationally modified complexes allowed us to gain insight into the effects that the addition of PTMs has on the stability of protein-protein binding (S2 Table). The same analysis was also performed on the non-modified complexes, which served as a control. Notably, a total of four systems (1VLU, 2EKE, 4DL0, and 4WXA) were excluded from all subsequent binding energy analyses because the changes that took place during their simulations (e.g., dissociation of a dimeric complex) caused the violation of periodic boundary conditions, i.e., proteins could "feel" their respective periodic images; these systems can easily be identified as those showing extreme RMSD values (S3 Fig; discussed below).

Similarly to our previous results on a smaller dataset [24], the comparison of the binding energy decomposition between non-modified and modified complexes revealed that lysine acetylation more frequently contributes to binding in a locally stabilizing fashion, while the opposite is true for phosphorylations (Figs 2 and S4). These effects might be the consequence of the local changes introduced by the respective modifications: while phosphorylation brings about a negative charge that must be compensated by neighboring positively charged residues,

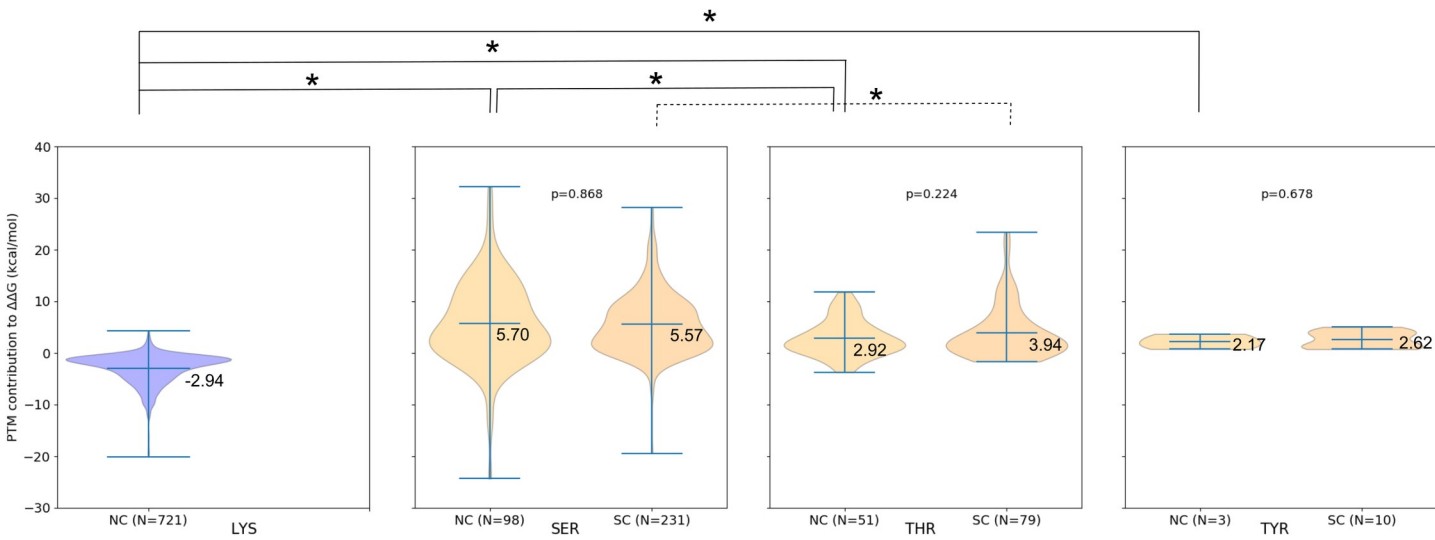

**Fig 2. Comparison of local binding contributions between conditions and PTM types for the interface-located PTMs.** $\Delta\Delta G_{bind,contribution}$ values in two conditions for each type of PTM (acetylation in blue, phosphorylation in orange), where NC stands for "normal conditions" and SC for "stress conditions". Acetylation more frequently contributes to binding in a stabilizing way, as reflected in the majority of the violin plot being below zero, while the opposite is true for phosphorylation. No significant differences were found between $\Delta\Delta G_{bind,contribution}$ for PTMs of the same type between normal and stress conditions (*p*-values indicated on the plots). Comparison of $\Delta\Delta G_{bind,contribution}$ values between different types of PTMs within a given condition shows several statistically significant differences, connected by the lines (full line for NC and dashed for SC) and marked with stars above the plots. The corresponding *p*-values are: *p*(Lys-Ser, NC) = 1.3e-76, *p*(Lys-Thr, NC) = 7.7e-39, *p*(Lys-Tyr, NC) = 2.3e-03, *p*(Ser-Thr, NC) = 2.0e-02, and *p*(Ser-Thr, SC) = 2.9e-02. Cohen's *d*-values indicate that the effect sizes vary from small (*d*(Ser-Thr, SC) = 0.29, *d*(Ser-Thr, NC) = 0.45) to large (*d*(Lys-Ser, NC) = 1.43, *d*(Lys-Thr, NC) = 1.82, *d*(Lys-Tyr, NC) = 2.23).

acetylation diminishes the charge from the lysine side chain. Notably, these local contributions of PTMs do not necessarily dictate the overall impact on binding affinity, which we observed here, as well as in our previous work. For instance, while the local effect of phosphorylation might be destabilizing, the overall effect on the binding of protein subunits can be stabilizing because there are other effects, such as conformational changes, also taking place (discussed below).

Focusing on the PTMs located at the binding interfaces, this difference between acetylation and phosphorylation is statistically significant (*p*-value < 0.05) with a large effect size (Cohen's *d*-value > 0.8) when comparing lysine acetylation $\Delta\Delta G_{bind,contribution}$ values with those for serine, threonine or tyrosine phosphorylation (Fig 2). Moreover, serine phosphorylation appears to have a significantly larger destabilizing effect on the binding when compared to phosphorylation of threonine in both normal and stress conditions, though the effect sizes are small. When $\Delta\Delta G_{bind,contribution}$ values are compared for the same type of PTM between normal and stress conditions, as expected, no statistically significant differences are found (Fig 2).

## Co-occurring PTMs together affect protein interactions

While the previous section describes the local contributions of PTMs to the binding, the overall effect that multiple PTMs co-occurring within a single protein complex exert onto binding ($\Delta\Delta G_{bind}$) is more complicated than a simple sum of these local contributions (Fig 3). This makes straightforward predictions based on multiple local effects impossible and is due to PTMs causing conformational changes (i.e., they work through allostery). Furthermore, we have observed a significant difference (*p* = 5.9e-05) in distributions of $\Delta\Delta G_{bind}$ values for normal and stress conditions, where values are more stabilizing for normal conditions (Fig 4). Cohen's *d* value of 0.32, corresponding to the difference of 0.32 standard deviations, indicates that the observed effect size is not large. The observed difference may in part be due to the variation in the number of specific PTM types in normal vs. stress conditions, i.e., all acetylations in our

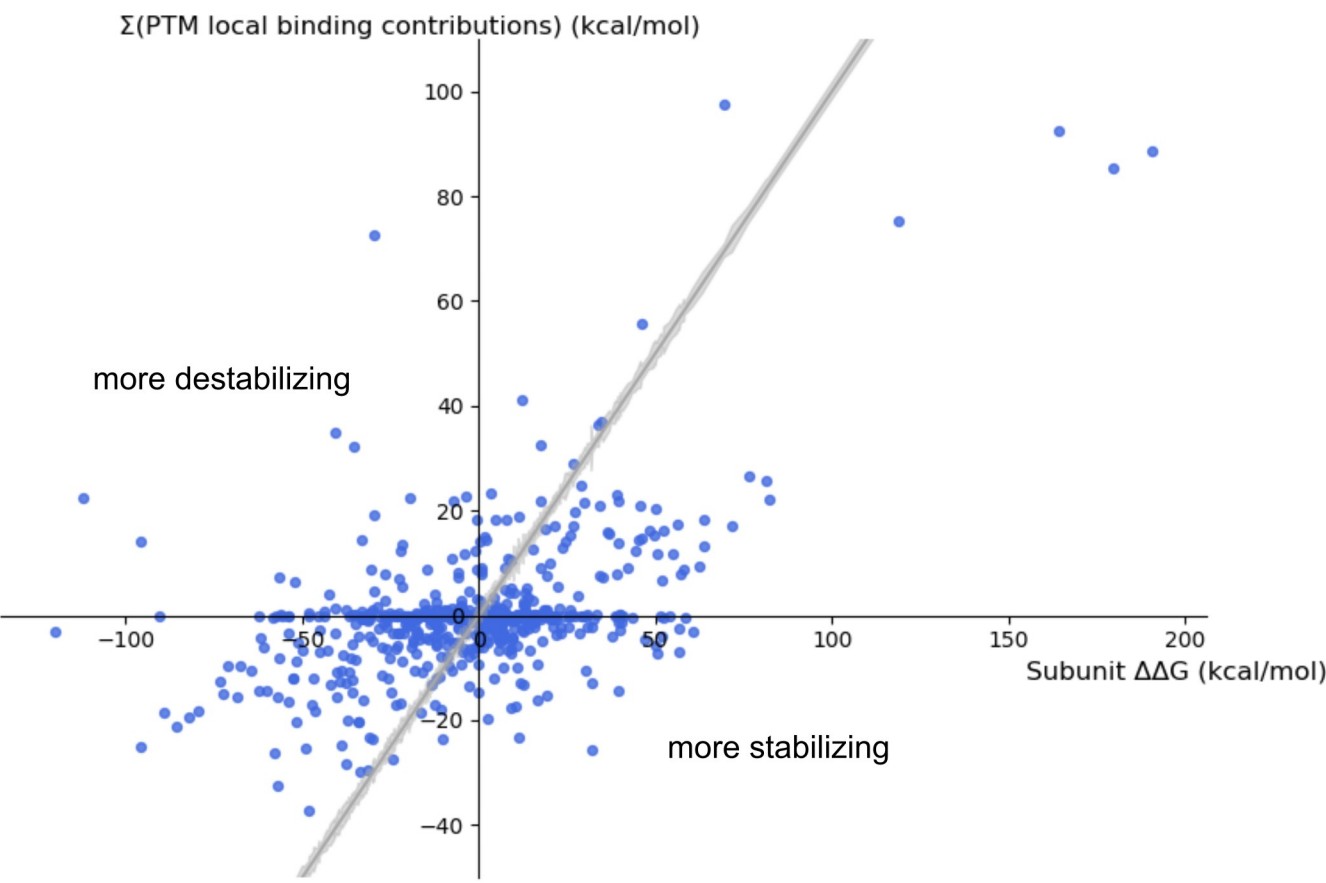

**Fig 3. Correlation between the subunit $\Delta\Delta G_{bind}$ values and the sum of $\Delta\Delta G_{bind,contribution}$ of PTMs located in the respective protein chains.** Points above the grey line ($y = x$, with the surrounding grey area describing the error in subunit $\Delta\Delta G_{bind}$ values) belong to systems in which the overall effect of PTMs on subunits' binding is more destabilizing than could be expected based solely on local contributions of the PTMs, and the other way around for points below the grey line, both thanks to long-range effects. For systems described by points lying within the grey area, the local effect of PTMs does explain the overall effect on binding.

dataset were identified in normal conditions, while overall a larger number of phosphorylations was assigned to stress conditions. However, on a level of individual complexes with PTMs identified in both conditions, $\Delta\Delta G_{bind,normal} < \Delta\Delta G_{bind,stress}$ does not always hold true (S5 Fig), further indicating complex pathways in which co-occurring PTMs exert their effect on binding.

The overall effect of PTMs onto binding might be the consequence of changes in the bound state, the unbound state, or both, where the bound state refers to the entire protein complex and unbound to the two components during the MM/GBSA calculations, which can be thought of as a receptor and a ligand. Estimation of their contributions to the overall $\Delta\Delta G_{bind}$ (S2 Table) indicates that the contributions vary from one system to the next. On average, the unbound state has a somewhat higher contribution (S6 Fig; $p$ = 9.5e-23), although the effect is only of a medium size (Cohen's $d$ = 0.55). A caveat that should be noted here is that the unbound states were inferred from the MD simulations of the complexes, rather than originating from independent simulations of the individual protein components.

## Protein conformational changes due to PTMs

We used root mean square deviation (RMSD) to estimate the conformational changes in the analyzed protein complexes throughout the MD production phase, as compared to their

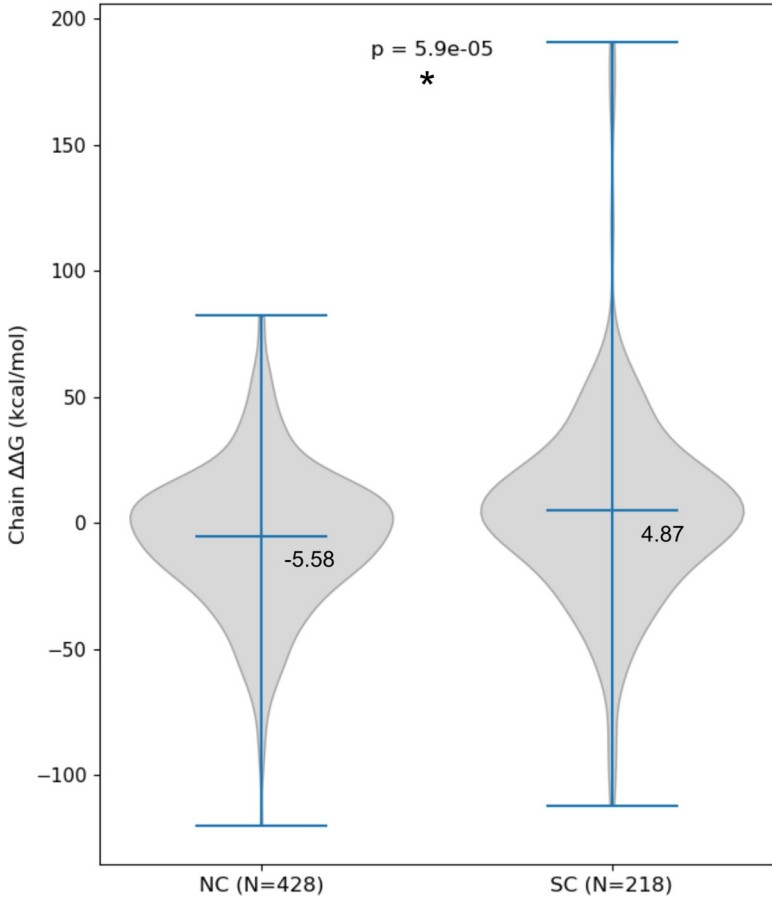

**Fig 4. Overall effect of PTMs on the protein-protein binding.** Chain $\Delta\Delta G_{bind}$ values are significantly more stabilizing for PTMs found in normal when compared to those identified in stress conditions, though the effect size is small (Cohen's $d$ = 0.32).

respective initial structures (i.e., before optimization). Because these initial conformations are the same for both non-modified and modified complex and correspond to the structure reported in PDB, these RMSD calculations describe how much the conformations for non-modified and modified complexes acquired during MD simulations deviated from the initial one (S3 Fig). The average value of RMSD for the majority of protein complexes went up to 5 Å, both when calculated for backbone or entire proteins, with some of the more extreme cases going above 10 Å (S3 Table). As already mentioned, the outlying systems with extreme RMSD values are 1VLU, 2EKE, 4DL0, and 4WXA, in which the violation of periodic boundary conditions was observed. The underlying changes that occurred during the MD simulations and have caused such results are dissociation in one (modified 1VLU; non-modified 4WXA) or all simulated systems (both non-modified and modified 2EKE), or a significant conformational change as compared to the initial, crystal structure (4DL0). RMSD values for these systems are reported in the Supplemental material, however they are excluded from RMSD analyses detailed below.

When comparing the RMSD of the modified proteins and their non-modified counterparts, a large number of systems showed a difference of less than 1 Å (the difference of areas under the RMSD graphs divided by time; value similar to the difference of the average RMSDs; S7A Fig). In some cases, this difference was up to several angstroms, and depending on the system

it was either the non-modified or modified version of the complex that had a larger RMSD. In general, more systems showed larger conformational changes for modified than for non-modified complex (149 out of 255 when comparing backbones, and 162 out of 255 when comparing entire proteins), and we find no significant difference between normal and stress conditions ($p_{backbone}$ = 0.87). The slight difference between results for the backbone and entire protein suggest that there are systems in which the overall shape (backbone) of the modified complex stays more similar to the starting conformation than the non-modified one; however, if side chains are also taken into account, they add enough of a difference to make the modified complex more conformationally distant from the initial structure than the non-modified one. Thus, we find that the protein complexes during MD typically do acquire conformations rather distant from the initial one, with modified complexes more frequently showing larger changes than their non-modified counterparts, as might be expected given that all the initial conformations originate from presumably non-modified versions of protein complexes.

While the above-described differences in RMSD between modified and non-modified complexes are typically small, suggesting that both versions of the complex became approximately equally distant from the starting conformation, this does not imply that they acquired a mutually highly similar conformation. To directly assess the difference of conformations between non-modified and modified complex, RMSD between the two was calculated. In order to find the representative structures for this comparison, clustering of conformational snapshots from the MD production phase was performed and the representative structure of the largest cluster was then taken for the RMSD calculation. The obtained values were typically between 1 and 2 Å, and went up to 10 Å for some systems (S7B Fig). Overall, the higher RMSD values between the cluster representatives suggest that the modified and non-modified complex typically do differ conformation-wise more than could be expected based on their distances from the initial conformation.

## PTM sites do not show increased conservation over their non-modified counterparts

Thus far, the reports on whether PTM sites are more conserved than the non-modified amino acids have been contradicting.[9] A unified view on this problem has been suggested for phosphorylation: phosphosites show a slightly increased conservation, but only when compared to non-modified amino acids of the same type that are located in the same type of secondary structure region (ordered or disordered) of the same protein.[9] Similarly, lysine PTM sites were also found to be only slightly more conserved when compared to non-modified lysines. We have therefore taken a closer look into the location of PTM sites from our dataset within different secondary structure elements. While acetylation sites of our dataset follow a similar secondary structure pattern as the non-modified lysines and are mirroring the overall distribution of secondary structure elements across the investigated protein structures, phosphorylation sites showed a clear preference for the unstructured elements (S8 Fig). However, even with taking the secondary structure into account, we found no statistically significant differences in conservation between modified and non-modified amino acids (S9A Fig and S2 Table). In addition, we do not find a significant difference between the conservation of phosphorylation and acetylation sites (S9B Fig).

Finally, one might expect that the more conserved residues will also have a larger effect on the protein-protein binding. However, we find no correlation between $\Delta\Delta G_{bind,contribution}$ values and conservation of PTM sites (S10 Fig), either when taking all or only the subset of interface-located PTMs into account, where the interface-located PTMs can be identified as those with zero $\Delta\Delta G_{bind,contribution}$. It could be argued that this is due to complex mechanisms by

which PTMs exert their effect on protein binding, i.e., individual local effects have impact, but do not determine, how the overall protein-protein binding will be affected, and this is especially true for proteins with co-occurring PTMs. Thus, we do not find that conserved residues have a larger effect on protein-protein binding than non-conserved sites.

## Mirroring results to organisms beyond yeast

It could be argued that predictions of PTMs' effect on protein binding in yeast can be mirrored to other organisms, given that they have the corresponding orthologous proteins and PTMs within them conserved. To assess how widely the PTMs of our dataset are conserved, for each PTM we constructed a comprehensive list of organisms in which the respective PTM site can be found (S2 Table). These lists were then used to search for the lowest common ancestor of organisms with conserved PTMs for each PTM site. The majority of PTMs indeed appears to be conserved rather widely (Fig 5). Furthermore, a table describing the proportion of conserved PTM sites per protein or PDB structure was assembled (S3 Table). This data provides a detailed overview of which predictions from yeast might be mirrored to which other organisms.

## A case study–importin alpha

To highlight how the data obtained within this work can be used and interpreted, we took a closer look at one of the systems–importin or karyopherin alpha (Kapα, PDB ID: 1BK5), which has associated PTMs in normal conditions. The role of this protein within the yeast cell is connected to the nuclear import: Kapα binds the positively charged nuclear localization signals (NLS) in the cytosolic proteins, directing them for transport into the nucleus, and this recognition is enhanced when Kapα is in a heterodimer with Kapβ. In the suggested mechanism (S11 Fig), Kapβ binding activates Kapα by releasing the auto-inhibition, which is caused by binding of an internal Kapα NLS motif to Kapα itself.[28] Co-crystallization of Kapα with an external NLS motif (PDB ID: 1BK6) uncovered two binding sites within this protein, consistent with it being able to bind both single and bipartite NLSs.[29] Moreover, it was found that Kapα forms a homodimer through interactions of the C-terminal region (PDB ID: 1BK5), forming a substantial interface which buries 13% of each subunit's surface. Although this finding is seemingly incompatible with bipartite NLS binding, as the homodimer interface is located between two binding sites within a Kapα subunit, it was suggested that dimer formation represents yet another layer of the auto-inhibition that has to be surpassed by Kapβ binding.

A total of 6 lysine acetylations and 1 serine phosphorylation site were identified within Kapα in normal conditions. Conservation of these PTMs varies from low/medium (Lys479 (0.03), Lys496 (0.05), Lys398 (0.45)) to high (Ser351 (0.72), Lys485 (0.83), Lys500 (0.86), Lys354 (0.91)), with all of them being widely conserved. While none of these PTMs overlaps with NLS binding sites, mapping on 3D structure clearly shows their overrepresentation around the homodimer interface (Fig 6A). In this work, we predict that the addition of these PTMs works towards significant stabilization of Kapα:Kapα binding, as suggested by the negative values of $\Delta\Delta G_{bind}$ (S2 Table) On the local level, acetylations were predicted to contribute to binding in a stabilizing manner, while phosphorylation had a small destabilizing contribution, with only a subset of PTMs being located on the interface ($|\Delta\Delta G_{bind,contribution}| > 0.5$ kcal/mol). In addition, the search for the amino acids that have significantly different $\Delta\Delta G_{bind}$ contributions between non-modified and modified complexes revealed eight residues, both modified and non-modified, with contributions larger than 5 kcal/mol (S2 Table and Fig 6B). All of them were stabilizing, suggesting that PTMs addition to Kapα works towards

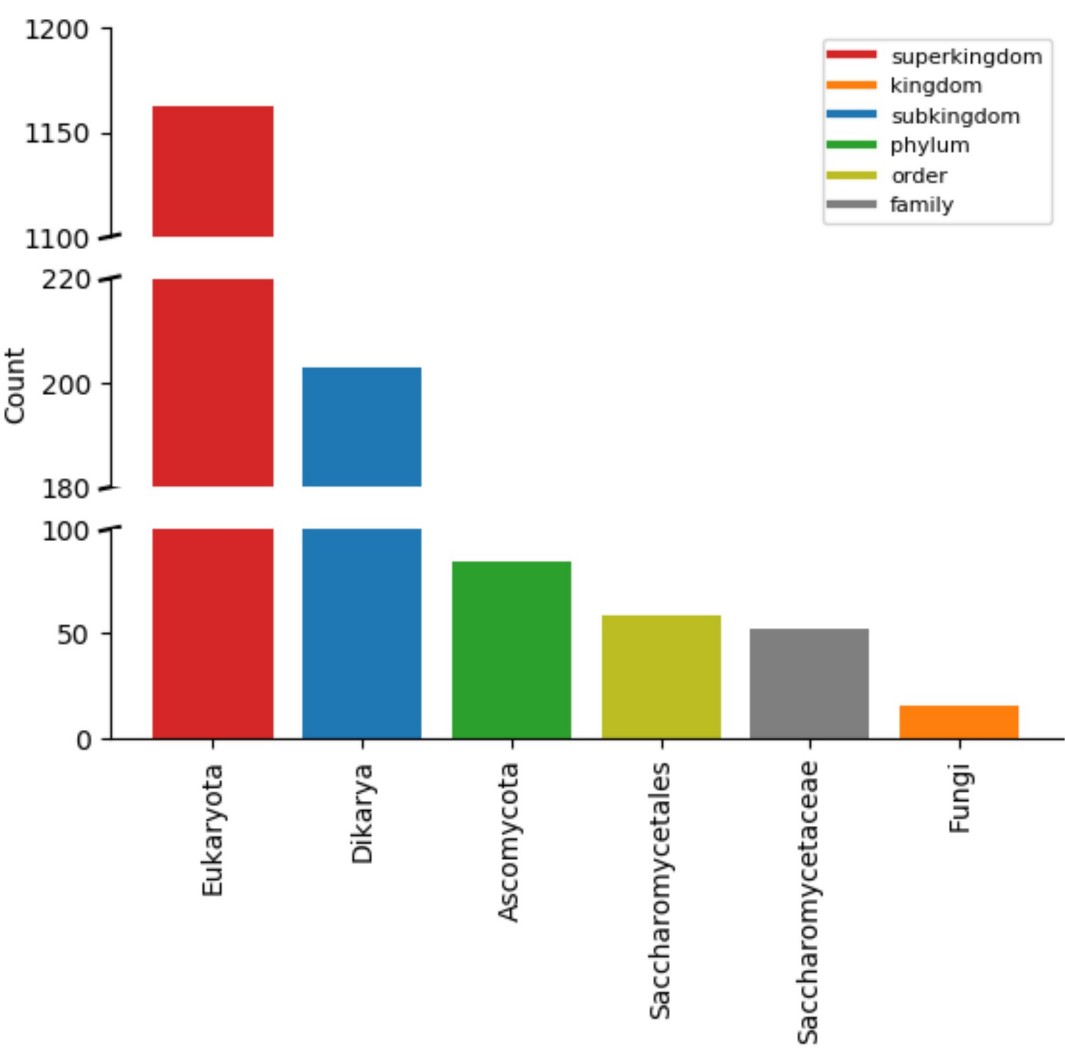

**Fig 5. Conservation of PTM sites.** PTM sites in the dataset used in this work appear to be widely conserved, with Eukaryota being the most common lowest common ancestor. The plot shows data for all unique PTM sites analyzed in this work, both acetylation and phosphorylation and independent of the conditions (normal or stress), where *unique* means that the redundancy due to PTMs appearing in multiple chains, complexes, or in both conditions was removed.

homodimer stabilization both directly (through PTMs) and indirectly (through other interface residues).

A detailed look into MD trajectories of the non-modified and modified Kapα dimer pointed towards different ways in which the residues with large contributions achieve their stabilizing effect upon PTM addition. It must be noted, though, that these are one-time observations originating from only a single MD simulation performed for each system, as well as that the longer simulation time would be preferable for such analyses. With these caveats in mind, our analysis showed that the conformational changes caused by the introduction of PTMs can relocate a residue from a non-interface to an interface position, where it then forms favorable interactions with the opposite chain. One such example is Glu460 in chain A, which was observed to form a salt bridge with Lys375 of the same chain in Kapα$_2$. However, in the modified complex, this Glu460 moves towards the dimer interface, where it begins forming a salt bridge with Arg120 of chain B (Fig 6C). Similarly, Glu493 in chain B only forms interactions

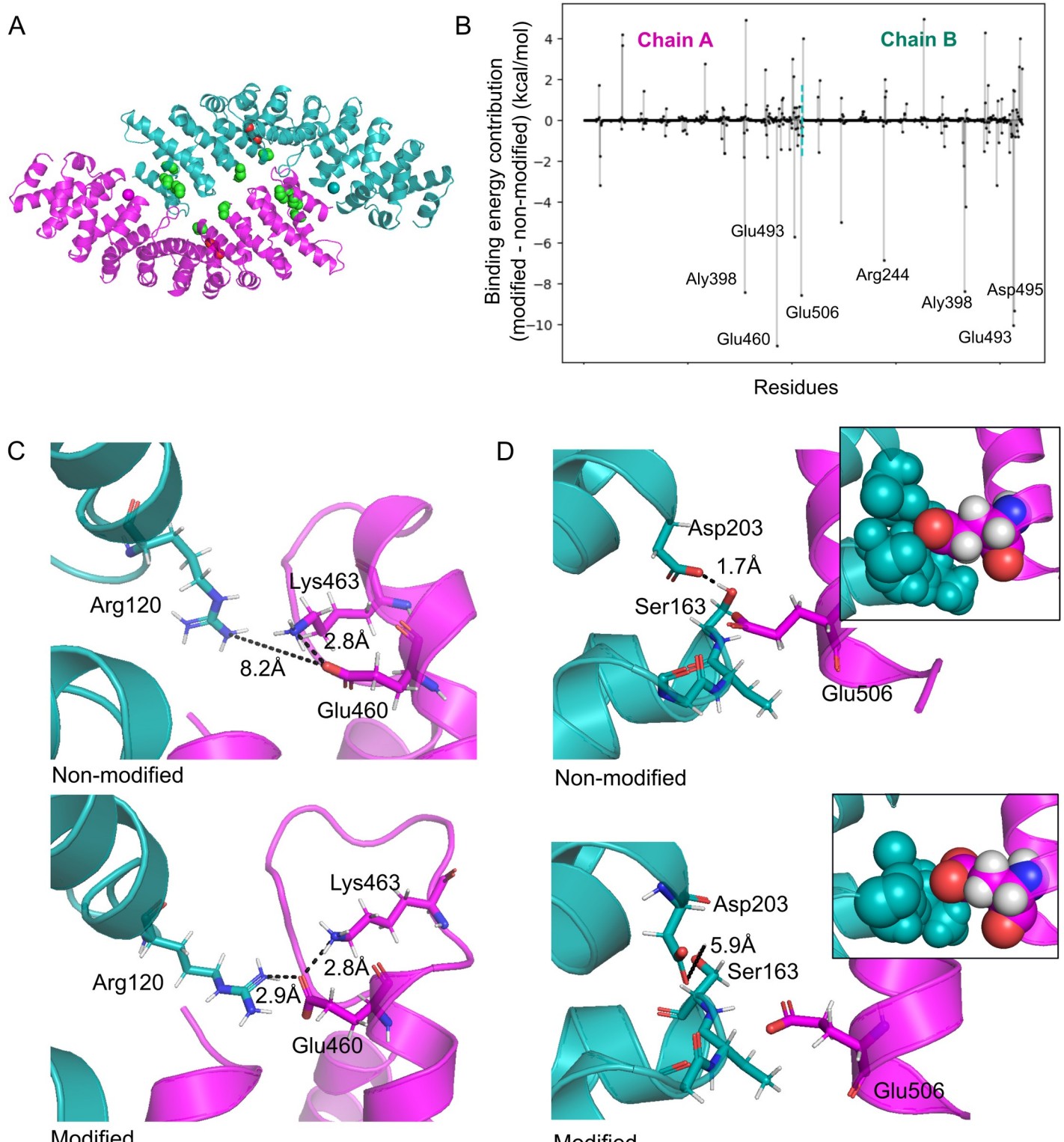

**Fig 6. A case study–importin alpha. A.** The initial structure of the Kapα homodimer (PDB ID: 1BK5), with chain A shown in magenta and chain B in cyan. Green space-filling representation is used to denote the backbone atoms of lysine acetylation sites, while red is used for serine phosphorylation sites detected in normal conditions. **B.** Per-residue decomposition of the Kapα free energy of binding ($\Delta\Delta G_{bind,contribution}$). Amino acids of homodimer are shown on the $x$-axis, where the two chains are separated by a dashed cyan line. For each residue, the difference of binding contribution between modified and non-modified complex is shown as a vertical line ending with a dot, where negative values denote residues with a more stabilizing contribution in the modified than in the non-modified complex, and vice versa

for the positive. Residues with contributions larger than 5 kcal/mol are labeled. **C.** Glu460 in chain A is interface located only in the modified complex, and therefore has a large stabilizing $\Delta\Delta G_{bind,contribution}$, mainly due to interactions with Arg120 from chain B. **D.** Glu506 has a significantly less destabilizing contribution to binding in the modified complex, where chain B is more distant. The inserted frames depict Glu506 and residues of chain B which are within 5 Å from it in a space-filling representation.

(salt bridges, hydrogen bonds) with chain A in the modified complex. Different from this simple scenario, the addition of PTMs can also cause more complex changes in amino acid interaction patterns, ultimately leading to highly stabilizing binding contributions for some residues. For example, the loop 160–163 in chain B moves further away from the helix in chain A when Kapα₂ is modified. This rearrangement includes changes in the interaction pattern, such as the hydrogen bond between Ser163 and Asp203 of chain B, which is lost in the modified complex. Because the aforementioned loop contains no residues that could form highly favorable/stabilizing interactions with Glu506 in the opposing helix of chain A, this negatively charged residue has a significantly less destabilizing contribution to binding in the modified homodimer (Fig 6D). In another example, the stabilizing Glu493 in chain A weakly interacts with Arg321 of the opposite chain, as this arginine is also involved in an interaction with Glu360 of the same chain. Upon PTMs addition, Glu493 positions in the proximity of Lys359 of chain B, forming a stable salt bridge and consequently stabilizing the binding more than in the non-modified complex. Taken together, this kind of analysis might help in shedding more light onto the roles of PTMs in Kapα₂ binding, and moreover the possible role of PTMs in the broader context of homodimer formation as a mechanism of auto-inhibition.

## Discussion

PTMs play vital roles in virtually all cellular processes, yet studying their impact on proteins using molecular modeling and MD simulations, known to be valuable tools for understanding the dynamic nature of biomolecules, became possible only recently due to technical advancements, such as the development of parameters for describing the modified amino acids.[30] Starting from only a few common modifications, such as phosphorylation and methylation, the parameters have now been developed for over a hundred types of modified amino acids and are available in several MD suites.[31] In addition, specialized web-servers, such as Vienna-PTM [32] and FF_PTM [33], were developed for the addition of PTMs onto 3D protein structures and their parametrization. The application of MD simulations on studying PTMs often focuses on understanding how changes in protein structure are linked to the functional roles of PTMs. For instance, MD simulations of the N-terminal domain of the PLB protein showed that phosphorylation of Ser16 decreases the helical content of this protein part, subsequently modulating its interaction with the sarcoplasmic reticulum Ca-ATPase, relieving its inhibition.[34] In another example, simulations of phosphorylated kinase-inducible domain revealed not only the stages of its folding upon KIX binding, but also the key amino acids involved in this process.[35] Some studies also go beyond the quantitative observation of protein conformational changes upon PTM addition and analyze modified protein's physicochemical properties. For example, MD has been used to investigate polar properties of solvent accessible surfaces of phosphorylated proteins, where it was found that phosphorylation decreases hydrophobicity around the phosphosite, while the overall effect on the entire surface of a protein cannot be easily predicted due to dynamic structural rearrangements.[36] Others have taken advantage of physicochemical parameters calculated based on MD simulations to predict the dynamic behavior of protein complexes depending on the pattern of the interface-located phosphosites.[37].

Building upon the mentioned technical developments, this work uses MD simulations of post-translationally modified proteins as part of the recently developed MD-MM/GBSA workflow [24], aiming to contribute to our understanding of the effects that PTMs exert on protein binding, including complex cases of multimeric and multiply modified complexes. Due to the dynamic approach, these simulations manage to grasp both short- and long-range effects that the introduced PTMs have on the protein complex. The simulations show that the changes on the local level are dominantly stabilizing for lysine acetylation and destabilizing for serine/threonine/tyrosine phosphorylation, with the observed pattern likely being the consequence of the local amino acid charge change. However, it remains less clear why serine phosphorylation apparently has a more destabilizing effect compared to the same PTM on a threonine residue. While predictions of these local effects are seemingly straightforward, it is well established that PTMs can have a significant allosteric impact on protein structures. For instance, phosphorylations in the loops surrounding the peptide-binding site of Hsp70 chaperone were shown to be a part of the allosteric network that dictates the affinity towards this chaperone's substrates. [38] Moreover, similar evidence exists for lysine acetylation sites, e.g., cell-cycle dependent acetylation of the catalytic Lys33 of Cdk1 was demonstrated to have both short- and long-range effects, ultimately interfering with the cyclin-B binding through changing the corresponding binding interface.[39] The predictions of the effects of PTMs in this work go alongside these notions, as there are multiple cases of protein complexes in which the PTMs are positioned outside of the interaction interfaces, and yet they exert a non-zero effect on the free energy of binding between subunits of the complex (data points lying on the *x*-axis of Fig 3), clearly suggesting the existence, as well as the importance of the long-range effects of PTMs.

With the obtained dataset at hand, we have also tackled the question of overall conformational changes occurring in protein complexes due to PTMs introduction, both on the entire protein and backbone levels. The average change for both modified and non-modified complexes compared to the initial structure was at least 1.3 Å on the level of entire protein, and in over 75% of cases above 2 Å, in line with expectations after a 20 ns MD simulation. When comparing the conformations of the cluster representatives of non-modified and modified complex, the average RMSD was 2.2 Å, with 35% of systems (90 out of 255) going above 2 Å. A recent work by Xin and Radivojac has also dealt with the same problem of determining the amount of conformational change caused by PTMs. However, their RMSD calculations were based on static protein structures of non-modified proteins and their modified counterparts as reported in PDB, while focusing mainly on glycosylation and phosphorylation.[40] They reported large conformational changes (RMSD > 2 Å) in only 7% of glycosylated and 13% of phosphorylated proteins, which is somewhat lower compared to our findings, perhaps due to differences in types of studied modifications and/or underlying methodologies. Therefore, although in some cases the effect is subtle, the PTMs commonly do have an impact on the conformation of the affected protein, further confirming that the allostery is an important mechanism through which PTMs exert their functions.

A significant part of the proteome can contain co-occurring PTMs and this holds true for a number of organisms (Table 1).[18] An extreme example is the human serine/arginine repetitive matrix protein 2 (Srrm2), whose 2,752 amino acids long sequence has 281 PTM sites listed in the UniProt database. PTMs located within the same protein can influence each other, which is often referred to as the PTM crosstalk.[1,41] Depending on the amino acid type, different PTMs can compete for the same site and thereby modulate the function of a protein, e.g., lysine residues at the C-terminal domain of p53 can be either acetylated or ubiquitinated, with the type of modification dictating whether the respective protein will be targeted to proteasomal degradation or not.[42] Moreover, one or several co-occurring PTMs can affect the modification status of another PTM site within the protein, such as in the case of the Mef2

**Table 1. Overview of the PTMs count in the Swiss-Prot entries for human and model organisms.** [18].

| | Swiss-Prot entries | Entries with PTMs | Max N(PTMs) in an entry | N(entries) with given number of PTMs | | | | | | Modified entries with >1 PTM (%) |
|---|---|---|---|---|---|---|---|---|---|---|
| | | | | 1 | 2 | 3 | 4 | 5 | >5 | |
| *Escherichia coli* K12 | 4,518 | 341 | 12 (P0A9B2, P0A6Y8, P0A825) | 273 | 32 | 13 | 9 | 6 | 8 | 19.9 |
| *Drosophila melanogaster* | 3,603 | 897 | 88 (Q8INM3) | 304 | 170 | 103 | 81 | 55 | 184 | 66.1 |
| *Saccharomyces cerevisiae* S288c | 6,721 | 2,344 | 40 (P32944) | 983 | 474 | 288 | 168 | 127 | 304 | 58.1 |
| *Arabidopsis thaliana* | 15,952 | 2,688 | 42 (Q9FZA2) | 1,635 | 437 | 150 | 153 | 90 | 223 | 39.2 |
| *Mus musculus* | 17,038 | 8,712 | 264 (Q8BTI8) | 2,438 | 1,460 | 962 | 667 | 601 | 2,584 | 72.0 |
| *Homo sapiens* | 20,365 | 9,372 | 281 (Q9UQ35) | 2,630 | 1,560 | 1,008 | 739 | 616 | 2,819 | 71.9 |

transcription factors family, in which phosphorylation promotes the sumoylation of lysine in the same motif.[3] In addition, the specific combination of PTMs modifying a protein at a given moment can determine its function or interactors. For example, the combination of acetylated Lys381 and dimethylated Lys382 directs a large conformational change in p53, driving its binding to the Tudor domain of 53BP1 protein.[43] In addition, nearby phosphorylations of serine and threonine were found to further modulate p53:53BP1 interaction. The existence of the crosstalk, together with the knowledge that many proteins indeed contain multiple PTM sites, makes predictions of PTMs' effects on binding of a questionable biological relevance when they are done for a single PTM at a time. However, because many PTMs, including acetylation and phosphorylation, are reversible, introducing multiple modifications into a protein complex for the purpose of making a prediction introduces yet another difficulty, which is knowing exactly which combination of PTMs is present on a protein. While there are techniques that allow a more detailed view on cellular PTMs, such as top-down proteomics which identifies individual proteoforms [44] or methods that identify PTMs upon perturbations [45], typical identification of PTMs by MS is done from the cell lysates using bottom-up approach in which the fine level of detail is lost. Consequently, the bulk of the data stored in the public databases originates from the bottom-up studies. In addition, it should be kept in mind that the pool of the known PTMs is still increasing [17], rendering our current knowledge incomplete. As a compromise, in this work the PTMs from the publicly available databases were classified into those detected in normal or stress environment, based on the conditions in which the yeast was grown preceding the PTMs' detection. While this approach represents a certain simplification of the complex PTMs landscape, it could still be argued that it produces more meaningful predictions compared to those for individual PTMs.

Depending on the type, some PTMs appear to have a preference for being located in either ordered or disordered secondary structure elements.[46] While lysine acetylation has no apparent preference for secondary structure placement, serine/threonine/tyrosine phosphorylations were found to exhibit a preference for disordered regions. These secondary structure preferences were confirmed in this work for both of these respective PTMs (S8 Fig). Location within disordered regions has been suggested to have certain advantages: it might allow for faster reactions, as well as easier recognition of PTM sites by modification enzymes.[5,47] For phosphorylation, location in terms of the secondary structure has also been proposed as a factor to be taken into account when looking into phosphosites conservation.[9] In general, many types of PTMs, including acetylation and phosphorylation, appear to have only slightly higher conservation when compared to the equivalent non-modified residues. Two explanations have been proposed for this lack of strong PTM conservation [9]: i. a number of sites are not

functionally important and there is no evolutionary constraint to keep them, or ii. PTMs diverge while the function is preserved through other sites within the protein. Conservation of PTM sites has previously been suggested as a proxy for their functional importance [4], even though this measure cannot predict functional roles for those sites and it overlooks PTMs that are important only in their respective organisms [9]. In the context of protein-protein interactions, this work demonstrates that conservation of a PTM site is not a good predictor for the size of its local effect on $\Delta\Delta G_{bind}$. PTMs with low conservation and low $\Delta\Delta G_{bind,contribution}$ might: i. be located outside of the interaction interface and affect binding through long-range conformational changes; ii. work together with co-occurring PTMs (crosstalk), therefore exerting a common effect on binding, while the individual contribution stays small; or iii. have a role in another process which is unrelated to protein-protein interaction at hand. Even though the conservation of a single PTM appears not to be a good predictor for its local effect on the protein binding, it could be assumed that the predictions of the PTMs' effect on the binding of yeast proteins are transferable to other organisms, given that an organism has a set of conserved PTMs in an orthologous protein. This assumption makes results in the present work of a potentially even higher significance.

While the scientific community produces an ever-increasing amount of knowledge on both PTMs positions and protein structures for a number of organisms, predictions of PTM roles in protein interactions frequently overlook both their co-occurrence, as well as the dynamic nature of biomolecules. This work, therefore, represents a step towards creating a more realistic and complete picture of PTM roles in protein-protein interactions. Even though the focus of this work was placed on acetylation and phosphorylation in yeast proteins, the obtained results are potentially transferable to other organisms, and the presented methodology is applicable to making predictions in different protein complexes and for a wide array of modification types.

## Methods

### Post-translation modifications data

Proteome-wide lysine acetylation data for yeast *Saccharomyces cerevisiae* was downloaded from Protein Lysine Modification Database PLMD [15], containing 13,498 experimentally identified sites. A total of 16,551 serine, 3,835 threonine, and 371 tyrosine experimentally verified phosphorylation sites in yeast were retrieved from the dbPTM database [17]. The gathered PTM data originated from 25 different publications. Based on the conditions in which the yeast cells were grown, PTMs found in these studies were classified as being found in either *normal* or *stress* (e.g., DNA damage [48]) conditions.

### Selection of protein structures

At the time of retrieval, Protein Data Bank (PDB) [49] contained 3,425 *Saccharomyces cerevisiae* structures, 3,353 of which were successfully retrieved using Bio.PDB Biopython module [50]. For the remaining structures, the pdb file formats were not available in the database, typically due to their large sizes (e.g., containing ~80 chains, such as ribosome). Because performing molecular dynamics simulations with such structures would be very computationally expensive, this small subset of structures was left out from the further analyses. The successfully retrieved structures were then filtered based on a number of criteria. Firstly, they had to contain yeast, non-fusion protein chains that do not have engineered mutations reported in the pdb file. Secondly, with respect to the structure completeness, we distinguished the percentage of total protein length that was crystallized and the percentage of residues that were resolved in the structures (i.e., not in gaps), and required each of those to be at least 70% on

average. Furthermore, we kept only the structures with a resolution of 3.0 Å or better for the analysis (the criterium not applicable to NMR structures). Finally, membrane-embedded proteins were filtered out for the sake of simplicity, as well as all monomeric and structures without mapped PTMs, as the main point of interest of this work was the effect of PTMs on protein-protein interactions. When two or more structures in this reduced dataset represented the same protein complexes, the better structure based on resolution and completeness criteria was kept. The final dataset consisted of 179 structures: PTMs found only in normal conditions were mapped to 92 structures, PTMs identified only in stress conditions to 5, while the remaining 82 structures had two subsets of mapped PTMs discovered in these different conditions, respectively.

## Molecular dynamics simulations

For each protein structure, a molecular dynamics (MD) simulation was performed for fully modified complexes containing PTMs identified in normal and stress conditions, or only one of those depending on the available data. In addition, a simulation for a non-modified complex was ran as a control for each system. Overall, a total of 440 MD simulations were performed: 179 of non-modified complexes, 174 of complexes with PTMs found in normal, and 87 in stress conditions.

Preparation of protein structures and the MD runs were done as previously described.[24] The relevant biological assemblies from PDB were first cleaned from the solvent molecules (e.g., water, glycerol, ions), while the structural ions were kept in the proteins. All disulfide bonds detected by *pdb4amber* program of Amber16 molecular dynamics package [51] were kept in the structures. The addition of hydrogen atoms was done using *reduce* [52] from the same package. Non-resolved parts of the structures were detected based on residue numeration in pdb files and the amino acids flanking the gaps were capped with *N*-methyl (NME) as the C-terminal and acetyl group (ACE) as the N-terminal cap, using PyMOL [53]. PTMs were added to protein structures in an automated fashion, using a Python script (available in the Zenodo repository as PTMs_map) and the PyTMs [54] PyMOL plugin. Amber program *teLeap* was used for system parametrization, using *ff14sb* force field to describe proteins, *tip3p* for water, *phosaa10* [55] for phosphorylated residues, parameters from Khoury *et al.* [33] for acetylated residues, Joung/Cheatham parameters for monovalent, and Li/Merz parameters for 2 to 4 charged ions (12–6 normal usage set) optimized for TIP3P water, where ions were treated in a non-bonded fashion. Each system was neutralized by the addition of $Na^+$ or $Cl^-$ ions and surrounded by a box of explicit TIP3P water spanning 10 Å from the complex in each direction, therefore allowing for a minimum of 20 Å distance between the protein complex and its periodic image.

Optimization of structures was done in 25,000 steps divided into 5 cycles, each consisting of 1,000 steepest descent and 4,000 conjugated gradient steps, using *sander*. A constraint (force constant of 100 kcal/mol/Å$^2$) was applied to the entire protein in the first, protein heavy atoms in the second, and protein backbone in the third cycle. A lower (50 kcal/mol/Å$^2$) force constant was applied to backbone atoms in the fourth cycle, while no constraint was used in the final one. Following the optimization, each system was equilibrated during 500 ps with *pmemd*. Equilibration was done using 1 fs time step, SHAKE to apply constraint onto hydrogen-containing bonds, and a cutoff distance of 15 Å for non-bonded interactions. In the first stage of equilibration (0–300 ps), the canonical *NVT* ensemble was simulated using the constraint of 25 kcal/mol/Å$^2$ on protein atoms, and with temperature increasing from 0 to 300 K within the first 250 ps, followed by 50 ps of constant temperature. The isothermal-isobaric *NpT* ensemble was simulated in the remaining 200 ps of equilibration ($T$ = 300 K, $p$ = 1.0 bar)

with no constraints applied to the system. Finally, the production phase of 19.5 ns was done using Gromacs 5 software [56–63], with ParmEd 2.7 [64] used to convert the corresponding files from Amber to Gromacs format. For this final part of the MD simulation, the temperature was kept at 300 K using modified Berendsen thermostat and the pressure at 1.0 bar with Parrinello-Rahman barostat. The coordinates were written each 1 ps, while the time step was 2 fs. The LINCS algorithm was used to constrain hydrogen-containing bonds only. The cutoff distance of 12 Å was applied for non-bonded interactions, with the neighbor list being updated every 20 steps, while the particle mesh Ewald was used for the long-ranged electrostatic interactions. Simulations were done using the periodic boundary conditions and ran on the Flemish Supercomputer Centre (VSC) infrastructure.

Visual Molecular Dynamics (VMD) [65] was used for visualization of the trajectories. Gromacs functionalities were applied to correct trajectory for periodic boundary conditions, extract conformational snapshots for binding energy calculations, as well as RMSD calculations (detailed below). PyMOL was used to produce the figures containing protein structures, while the plots were made with Matplotlib [66].

## Binding energy calculation

Effect of PTMs addition on the binding of subunits within a complex was assessed by calculating the free energy of binding with MM/GBSA method (Molecular Mechanics energies with Generalized Born and Surface Area continuum solvation) using Amber MMPBSA.py.MPI [67], as was previously described and benchmarked [24]. Briefly, the following set of equations is used for a hypothetical AB dimer:

$$G = E_{bonded} + E_{electrostatic} + E_{vdW} + G_{polar} + G_{non-polar} - TS \qquad \text{(Eq 1)}$$

$$\Delta G_{\text{bind}} = \langle G_{\text{AB}} - G_{\text{A}} - G_{\text{B}} \rangle_{\text{AB}} \qquad \text{(Eq 2)}$$

$$\Delta\Delta G_{\text{bind}} = \Delta G_{\text{bind,ABmodified}} - \Delta G_{\text{bind,ABnon-modified}} \qquad \text{(Eq 3)}$$

The energy terms in Eq 1 are calculated with molecular mechanics, the polar solvation with generalized Born method, the non-polar solvation from a liner relation to the solvent accessible surface area, while the entropy term is oftentimes omitted [24,68], including in this work. We used 100 conformational snapshots from equally spaced time points from the final 10 ns of each MD trajectory to calculate $\Delta G_{\text{bind}}$ in Eq 2, where the needed topology files free of solvent were prepared using Amber's *ante-MMPBSA.py*. A single $\Delta G_{\text{bind}}$ value was obtained for dimers, while binding of each subunit (ligand) to the remainder of complex (receptor) was calculated for multimers, resulting in $n$ $\Delta G_{\text{bind}}$ values for an $n$-meric complex. In order to assess the effect that introduction of PTMs to the complex has on their binding, the difference between modified and non-modified complex is calculated as $\Delta\Delta G_{\text{bind}}$ according to the Eq 3. The main indicator when predicting the effect of PTMs is the sign of $\Delta\Delta G_{\text{bind}}$, with negative indicating stabilizing and positive destabilizing effects, and not the size of the $\Delta\Delta G_{\text{bind}}$ value. In addition, with *MMPBSA.py.MPI* it is possible to decompose the overall $\Delta\Delta G_{\text{bind}}$ on per-residue contributions in order to evaluate the impact that the individual interface-located amino acids have on binding in non-modified vs. modified complex. Throughout this work, a residue with a $|\Delta\Delta G_{\text{bind}}|$ contribution larger than 0.5 kcal/mol was considered as an interface-located one.

Finally, free energy values ($G$) for the complex, ligand, and receptor (i.e., the three terms in Eq 2) were obtained in each of the MM/GBSA calculations. Those values were then used as a starting point for estimation of the contribution of changes in bound (complex) and unbound

(ligand and receptor) states to the overall binding effect observed upon PTMs addition to the complex (i.e., $\Delta\Delta G_{bind}$). To that end, $\Delta G$ was calculated for each of those three molecular components from the respective free energy values for modified and non-modified system, e.g., $\Delta G_{complex} = \Delta G_{complex,modified} - G_{complex,non-modified}$. To estimate how much (percentage) the changes in each of the components contributed to the $\Delta\Delta G_{bind}$, each of the $\Delta G$ values was divided by the sum of absolute values of the three $\Delta G$ values.

## Conformational changes

In order to get insight into the size of the conformational changes in modified and non-modified complexes throughout MD simulations, we compared conformations from the trajectory corrected for periodic boundary conditions to the initial protein structure (the respective protein structure which was the output of *teLeap*). This provided a good reference point to compare modified complex with its corresponding non-modified version, as initial structures differed only in the presence of PTMs, but not in the conformation. Root mean square deviation (RMSD) calculation was performed using Gromacs package, with either "backbone" or "protein" (or "protein_PTMs") selected for the respective analysis, in order to estimate both overall and changes protein backbone. Conformational changes for the duration of the MD production phase were then compared between modified and non-modified complex by subtracting the areas under the RMSD graphs (modified minus non-modified) and dividing by the duration of the MD production phase. Systems with extreme RMSD values (1VLU, 2EKE, 4DL0, 4WXA) were checked for violation of the periodic boundary conditions, i.e., a situation in which a system comes close enough to its periodic image so that two can interact. Such calculations were performed using *gmx mindist* from Gromacs package with employing the *-pi* flag, respectively.

In order to directly compare conformations of the non-modified and modified protein, 100 equally spaced conformational snapshots from the last 10 ns of the MD production phase were used to perform clustering of the entire proteins, using Gromacs (*gromos* method and cutoff 0.25). RMSD was then calculated between the representative structures of the largest clusters for non-modified and modified complex, using *align* in PyMOL.

## Conservation analysis

The conservation of sites of interest was calculated from the EggNOG [69] alignment of the orthologous proteins, where the opisthokonts group was chosen, as it is the smallest group that includes both *Saccharomyces cerevisiae* and *Homo sapiens*. A lysine acetylation site is considered to be conserved if an orthologous protein contains lysine at the same location in the alignment as the yeast protein, or one position up/downstream. This is in agreement with the previous research which showed that despite acetylation sites not being strictly conserved, the +/-1 variation in the position likely allows them to keep the functionality.[12] A serine phosphorylation site was considered conserved if either a serine or a threonine residue appeared at the exact position in the alignment, and the same was true for threonine phosphorylation sites. Finally, a strict positional and amino acid conservation was required when it came to tyrosine phosphorylation sites. Unpaired *t*-test was applied to assess whether there is a significant difference between conservation of modified and non-modified residues.

Where secondary structure was taken into account, the assignment of individual residues was performed by DSSP [70] using Bio.PDB.DSSP from Biopython. For the sake of simplicity within this work, the secondary structure elements were defined as a helix (H; includes H, G and I), a sheet (E; includes E and B), or non-structured (C; includes S, T, C and -).

In order to assess which of the results obtained for yeast complexes might be transferable to other organisms, i.e., which organisms have proteins orthologous to yeast proteins with PTM sites conserved, a comprehensive list of organisms in which any given PTM site was conserved was assembled using *taxonomy* tool of the UniPept resource [71]. Finally, *taxa2lca* from the same resource was used to find the lowest common ancestor in which each of the PTM sites is conserved.

## Supporting information

**S1 Table. List of lysine acetylation and serine/threonine/tyrosine phosphorylation sites found in either normal or stress conditions in the respective chains of the selected *Saccharomyces cerevisiae* PDB structures.**
(XLSX)

**S2 Table. Binding energy calculation results on subunit and local level.** Local contribution to protein binding, secondary structure element assignment, and details of the conservation calculation are given for each PTM added to a protein complex in either normal or stress conditions. Moreover, the overall effect on binding ($\Delta\Delta G_{bind}$ values) is reported, as well as estimated contributions of changes in bound and unbound states. Finally, a selected set of local contributions for the case study system Kapα are shown.
(XLSX)

**S3 Table. Percentage of PTMs studied in yeast in this work that are also conserved in another organisms (0 = no PTM conserved between an organism and yeast; 1 = 100% of PTMs conserved), at the level of either PDB structure or protein.** In addition, information on conformational changes in modified complexes as compared to their non-modified counterparts, originating from either backbone or RMSD analysis of the entire protein, is reported.
(XLSX)

**S1 README. Description of scripts and files deposited in Zenodo.**
(PDF)

**S1 Fig. Overview of sizes of yeast protein complexes analyzed in this study in terms of A. multimeric state and B. number of residues.**
(TIF)

**S2 Fig.** Distribution of the number of A. all PTM sites, B. lysine acetylation, C. serine phosphorylation, D. threonine phosphorylation, and E. tyrosine phosphorylation sites among unique protein chains in the analyzed dataset. The darker shade of color in each subplot denotes normal and lighter stress conditions.
(TIF)

**S3 Fig. Backbone RMSDs for all complexes throughout simulations with initial conformations serving as references.** Grey denotes non-modified complexes, red complexes with PTMs in normal conditions, and cyan with PTMs in stress conditions. While most complexes are rather stable, the notable exceptions belong to the following systems: 1VLU, 2EKE, 4DL0, and 4WXA.
(TIF)

**S4 Fig.** Local contributions of A. all PTMs and B. interface located sites (i.e., contributions above 0.5 kcal/mol or below -0.5 kcal/mol) to the overall binding ($\Delta\Delta G_{bind}$).
(TIF)

**S5 Fig. Comparison of $\Delta\Delta G_{\text{bind}}$ in normal and stress conditions for systems with PTMs identified in both.** Lines connect the subunit $\Delta\Delta G_{\text{bind}}$ values, where the color of the line denotes their relationship. For the sake of clarity, the data is split based on $|\Delta\Delta G_{\text{bind,NC}} - \Delta\Delta G_{\text{bind,SC}}|$: **A.** <10 kcal/mol, **B.** 10–20 kcal/mol, **C.** 20–50 kcal/mol, and **D.** >50 kcal/mol.
(TIF)

**S6 Fig. Estimation of the contribution that changes in the bound (complex) and unbound (receptor and ligand) states have on the overall binding energy difference observed upon addition of PTMs.**
(TIF)

**S7 Fig.** The difference of conformational changes in modified and non-modified protein complexes A. with initial structures used as reference points and B. when comparing representative structures of the largest conformational clusters in the final parts of trajectories. Data points for systems 1VLU, 2EKE, 4DL0, and 4WXA are excluded from the plots.
(TIF)

**S8 Fig.** Amino acids placement within secondary structure elements in protein structures. Distribution of A. PTM sites and B. non-modified amino acids of the given type is shown, as well as C. their difference. "All proteins" bars describe the distribution of all amino acids in the secondary structure elements for the entire protein dataset, and serve as a reference.
(TIF)

**S9 Fig.** Comparison of the conservation levels. A. Even with the secondary structure taken into account, the PTM sites do not appear to be more conserved than the equivalent non-modified amino acids. B. No significant difference is found in the conservation of acetylation and phosphorylation sites.
(TIF)

**S10 Fig. Correlation of PTM sites conservation and local $\Delta\Delta G_{\text{bind,contribution}}$.** No correlation is observed in either **A.** normal or **B.** stress conditions for any of the PTM types (different colors), however, it is possible that PTMs with small contributions still do affect binding through long-range conformational changes, which is not captured by $\Delta\Delta G_{\text{bind,contribution}}$.
(TIF)

**S11 Fig.** The role and regulation of importin alpha. **(a)** The auto-inhibition of importin alpha (α) was suggested to occur both by binding of the internal nuclear localization signal (NLS), as well as homodimerization. **(b)** Binding of importin beta (β) to importin alpha releases the auto-inhibition by disrupting the homodimerization and displacing the internal NLS. **(c)** Formation of the α:β heterodimer enhances recognition of NLSs in the cytosolic cargo proteins and their subsequent translocation to the nucleus. Based on Goldfarb et al. (2004) [28] Trends Cell Biol. and Conti et al. (1998) [29] Cell.
(TIF)

## Acknowledgments

The computational resources and services used in this work were provided by the VSC (Flemish Supercomputer Center), supported by The Research Foundation–Flanders (FWO) and the Flemish Government–department EWI.

## Author Contributions

**Conceptualization:** Nikolina Šoštarić, Vera van Noort.

**Formal analysis:** Nikolina Šoštarić.

**Funding acquisition:** Vera van Noort.

**Investigation:** Nikolina Šoštarić.

**Methodology:** Nikolina Šoštarić, Vera van Noort.

**Project administration:** Vera van Noort.

**Resources:** Vera van Noort.

**Software:** Nikolina Šoštarić.

**Supervision:** Vera van Noort.

**Visualization:** Nikolina Šoštarić.

**Writing – original draft:** Nikolina Šoštarić.

**Writing – review & editing:** Nikolina Šoštarić, Vera van Noort.

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
