## [Decision Letter · Decision Letter 0]

4 Dec 2020

Dear Ms Sostaric,

Thank you very much for submitting your manuscript "Molecular dynamics shows complex interplay and long-range effects of post-translational modifications in yeast protein interactions" for consideration at PLOS Computational Biology.

As with all papers reviewed by the journal, your manuscript was reviewed by members of the editorial board and by several independent reviewers. In light of the reviews (below this email), we would like to invite the resubmission of a significantly-revised version that takes into account the reviewers' comments.

The reviewers all had questions about the MD simulation protocol as well as some additional issues that should be addressed.

We cannot make any decision about publication until we have seen the revised manuscript and your response to the reviewers' comments. Your revised manuscript is also likely to be sent to reviewers for further evaluation.

Sincerely,

Roland L. Dunbrack Jr., Ph.D.

Associate Editor

PLOS Computational Biology

Arne Elofsson

Deputy Editor

PLOS Computational Biology

Reviewer's Responses to Questions

**Comments to the Authors:**

Reviewer #1: The authors use equilibration molecular dynamics (MD) simulations, combined by the free energy calculation using MM/GBSA method to study the impact of phosphorylation and acetylation on the structure, dynamics and thermodynamics of protein complexes. The authors demonstrate find that acetylation tends to have locally stabilizing roles, while the opposite is seen in the case for phosphorylation. Importantly, the authors show that the modifications away from the binding site may significantly contribute to binding due to their effect on protein’s structure. Given their ubiquity and biological importance, post-translational modifications are an extremely important and timely area of research and I find the authors’ contribution to be relevant and publishable in PLOS Computational Biology, provided the following comments have been adequately addressed.

Major comments

1. The authors find that the acetylation in general acts in a stabilizing fashion, while phosphorylation is destabilizing. It is, however, not clear whether this effect is due primarily to the changes in the bound state, the unbound state or both. The authors should find a way to quantitatively estimate these contributions separately.

2. In addition to a large-scale analysis of a high number of systems, the authors also focus on a case study – importin alpha. While the low level of sampling per complex (20 ns) for the large-scale analysis can be justified by the high number of systems studied and the general focus on just the thermodynamics of binding, I found the detailed analysis of importin alpha not to be justified at such a low level of sampling. Having a case study that illustrates the general principles captured by the large-scale analysis is a welcome idea, but in that case the level of sampling should be on at least the microsecond level to reach the standards in the field. The authors should extend their simulations of importin alpha to this level, and reanalyze their data.

3. The authors should provide a more extensive discussion of the potential deficiencies of the MM/GBSA method for free energy calculations and discuss what they would expect to obtain with more accurate methods.

Minor comments

1. L29. “Stressful” should be changed to “stress”

2. L453 should be “due to their large sizes”

3. 3. How were the 100 conformational shapshots from the final 10 ns of each MD trajectory that were used for free energy calculation chosen? At random or from equally spaced time points? This should be mentioned in the methods.

4. The authors should more exhaustively cite the available literature especially on MD simulations of PTMs and their impact on fundamental biological processes (e.g. Petrov et al. 2013, PLOS Computational Biology, e1003154; Paterlini, M. G. et al. Biophys. J. 2005, 88, 3243-3251; Polyansky AA et al, J Phys Chem Lett 3: 973–976, 2012.

Reviewer #2: Review is uploaded as an attachment.

Reviewer #3: This study addresses the field of PTMs and whether they stabilize or destabilize protein interactions.

Not many computational studies have attempted this.

The authors have previously (in [24]) studied 3 protein complexes with the same GB/SA protocol and validated their findings against their own experimental data.

Here, they now inspected 174 protein complexes having 2832 lysine acetylation and 289 phosphorylation PTMs.

All of these systems were subjected to the same MD protocol, which is quite impressive.

Much of this work has been conducted quite carefully, e.g. I liked their preparation of the data set and the

initial statistics of PTMs.

However, I am a bit sceptical about the reported deltaG numbers.

(1) Fig. 2 suggests that all phosphorylation-PTMs are destabilizing. I don't think this is generally true.

Why would then e.g. 14-3-3 domains always preferably bind to phosphorylated binding partners?

The unphosphorylated variants should always have more favorable binding energies, which is not the case.

I think the authors should critically reflect on whether they want to stick to their strict interpretation.

At least, the authors need to find a way to convince the reader that GB/SA works fine for phosphorylation PTMs.

(2) One potential source of error could in fact be the method used to compute delta-delta-G values (GB/SA).

Such calculations are very popular, yet this does not mean that their results to estimate PTM energies can

simply be trusted.

In

https://onlinelibrary.wiley.com/doi/full/10.1002/wcms.1448

the authors reviewed the ability of various computational approaches to compute protein protein binding affinities. In section 4.1, they comment on the abilities of the GB/SA method. Based on this, the method seems to work generally acceptable, at least as a scoring method.

However, there are also other reports. E.g.

https://www.ncbi.nlm.nih.gov/pmc/articles/PMC6453258/

recently studied the ability of FEP calculations and mm-GB/SA calculations to predict the effect of charge-changing sequence mutations at protein-protein interfaces.

Whereas FEP worked quite well (RMSE 1.23 kcal/mol for exposed positions and 1.79 kcal/mol for buried positions), mm-GB/SA gave larger deviations (1.50 kcal/mol and 2.80 kcal/mol), see table 2.

For cases showing small effects (table 3), FEP gave an R2 value of 0.39, whereas gave mm-GB/SA gave practically zero correlation of 0.06!

As a caveat of this study, one should add that the authors did not perform MD simulations in solvent (as was done in this manuscript), but simply performed a rotamer search while keeping the protein backbone fixed.

I am not aware if the accuracy of GB/SA calculations to predict delta_delta_G-values upon phosphorylation-PTMs

has been benchmarked sofar. A problem is likely the lack of suitable experimental benchmark data.

Thus, for the moment, I suggest that the authors should add a cautious statement in the discussion section noting that the accuracy of GB/SA calculations for PTMs is still a bit unclear. I did look at their previous paper [24]. Yet, I do not consider Y2H screen data as quantitative data.

(3) Only one MD simulation was performed here for each system.

Hence, the conformational reorientations discussed in lines 319++ are one-time observations.

It is unclear whether these are reproducable if the simulations were repeated starting with a different

random seem for assignment of velocities. Thus, a caveat should be added.

Further points:

(4) line 179++ the reported p-values are all extremely small, which does not fit to the overlaps seen in Fig. 2B.

Maybe this results from the large number of samples?

I suggest that the authors also compute Cohen's d values to estimate the effect size.

(5) line 235: An RMSD deviation of 10 A reflects a major conformational change such as unfolding

of a domain or unbinding. Stably folded domains do not show such high RMSD during 20 ns simulations.

(6) line 297: all values below -20 kcal/mol are clearly unrealistically low. Such extreme values are

"features" of the GB/SA method, and do not exist in nature. A comment should be added.

(7) line 363++: It is well-known that a small compact folded protein would show an RMSD of 1.5 to 2 A after

an MD simulation of 10 - 20 ns length.

(8) line 503: PME was used for the LONG-RANGED electrostatic interactions.

Further, the manuscript would benefit from a careful grammar revision. I would e.g. add dozens of direct and indirect articles throughout the text.

**Have all data underlying the figures and results presented in the manuscript been provided?**

Reviewer #1: Yes

Reviewer #2: Yes

Reviewer #3: Yes

PLOS authors have the option to publish the peer review history of their article (what does this mean?). If published, this will include your full peer review and any attached files.

Reviewer #1: No

Reviewer #2: No

Reviewer #3: No
---

## [Decision Letter · Decision Letter 1]

25 Mar 2021

Dear Ms Sostaric,

Thank you very much for submitting your manuscript "Molecular dynamics shows complex interplay and long-range effects of post-translational modifications in yeast protein interactions" for consideration at PLOS Computational Biology. As with all papers reviewed by the journal, your manuscript was reviewed by members of the editorial board and by several independent reviewers. The reviewers appreciated the attention to an important topic. Based on the reviews, we are likely to accept this manuscript for publication, providing that you modify the manuscript according to the review recommendations.

The second reviewer has some requests for comment and/or minor revision. Please respond to these.

Sincerely,

Roland L. Dunbrack Jr., Ph.D.

Associate Editor

PLOS Computational Biology

Arne Elofsson

Deputy Editor

PLOS Computational Biology

[LINK]

Reviewer's Responses to Questions

**Comments to the Authors:**

Reviewer #1: The authors have successfully addressed all of my comments.

Reviewer #2: Overall, the manuscript has improved significantly. The authors have added important information via figures. Still a few questions remain about the added figures and text.

Comments

1. Supplemental Figure S4, D: There does seem to be a difference between the normal and stress level in subfigure D compared to A,B,C. Does this have implications?

2. P246, “initial structures”: Just for clarification, the initial structures are the modified and non-modified structures respectively after energy minimization? So they are aligned on their respective structures, not on the same structure?

3. Supplemental Figure 6: Thank you for adding this figure. It provides a clear view on the stability of the simulated structures. Although an additional periodic boundary analysis might not be necessary for all, could the authors please show that the structures with extreme RMSDs actually do not violate the periodic boundary conditions? This would be insightful, for example, for PDB structures: 1VLU, 2EKE, 4DL0, and 4WXA.

4. P265, “Thus, we find that the protein complexes during MD typically do acquire conformations rather distant from the initial one, with modified complexes more frequently showing larger changes than their non-modified counterparts.” This finding does not seem surprising as the starting structures all originate from a non-modified form of the protein and, therefore, it can be expected that perturbation of the initial system via modifications can lead to larger deviations from the original structure compared to the unmodified protein system. Is this what the authors are trying to point out?

5. P274-275, “Notably, there are also a few outliers, such as the 1VLU system with RMSD value for cluster representatives larger than 20 Å.” Similar to comment 3, can the individual monomers interact with their respective periodic images in 1VLU? 20 Å seems to be a large distance when taking into account that the distance between periodic images is 20 Å in the setup of the simulation box.

6. P553: “using a Python script and the PyTMs”. Is this python script also being provided by the authors in the supplementary information? It could potentially be advantageous for the scientific community to publish the manuscript with the python script that adds PTMs to a protein system in an automated fashion. 

7. The individual simulations of the protein systems are still on the short side and have only been performed once, but if both of these properties are pointed out in the manuscript, the reader can take this into consideration.

Reviewer #3: The authors have appropriately addressed my points.

**Have all data underlying the figures and results presented in the manuscript been provided?**

Reviewer #1: Yes

Reviewer #2: Yes

Reviewer #3: Yes

PLOS authors have the option to publish the peer review history of their article (what does this mean?). If published, this will include your full peer review and any attached files.

Reviewer #1: No

Reviewer #2: No

Reviewer #3: No

Figure Files:

Data Requirements:

Reproducibility:

References:

---

## [Editor Report · Decision Letter 2]

21 Apr 2021

Dear Ms Sostaric,

We are pleased to inform you that your manuscript 'Molecular dynamics shows complex interplay and long-range effects of post-translational modifications in yeast protein interactions' has been provisionally accepted for publication in PLOS Computational Biology.

Best regards,

Roland L. Dunbrack Jr., Ph.D.

Associate Editor

PLOS Computational Biology

Arne Elofsson

Deputy Editor

PLOS Computational Biology

---

## [Editor Report · Acceptance letter]

10 May 2021

PCOMPBIOL-D-20-01903R2 

Molecular dynamics shows complex interplay and long-range effects of post-translational modifications in yeast protein interactions

Dear Dr Sostaric,

I am pleased to inform you that your manuscript has been formally accepted for publication in PLOS Computational Biology. Your manuscript is now with our production department and you will be notified of the publication date in due course.

With kind regards,

Katalin Szabo
